# THE MAGNITUDE VECTOR OF IMAGES

## ABSTRACT

The magnitude of a finite metric space is a recently-introduced invariant quantity. Despite beneficial theoretical and practical properties, such as a general utility for outlier detection, and a close connection to Laplace radial basis kernels, magnitude has received little attention by the machine learning community so far. In this work, we investigate the properties of magnitude on individual images, with each image forming its own metric space. We show that the known properties of outlier detection translate to edge detection in images and we give supporting theoretical justifications. In addition, we provide a proof of concept of its utility by using a novel *magnitude layer* to defend against adversarial attacks. Since naive magnitude calculations may be computationally prohibitive, we introduce an algorithm that leverages the regular structure of images to dramatically reduce the computational cost.

## 1 INTRODUCTION

The topology community has recently invested much effort in studying a newly introduced quantity called *magnitude* (Leinster, 2010). While it originates from category theory, where it can be seen as a generalization of Euler characteristics to metric spaces, the magnitude of a metric space is most intuitively understood as an attempt to measure the *effective size* of a metric space. As a descriptive scalar, this quantity extends the set of other well known descriptors such as the rank, diameter or dimension. However, unlike those descriptors, the properties of magnitude are not yet well understood.

Even though the metric space structure of dataset is of the utmost importance to understand e.g. the regularisation behaviour of classifiers, magnitude has received little attention by the machine learning community so far. However, it turns out that one can instead investigate the magnitude of each data point separately, considering each data instance as its own metric space. Following this line of thought, magnitude vectors were introduced as a way to characterise the contribution of each data sample to the overall magnitude, such that the sum of the elements of the magnitude vector amounts to the magnitude. As shown in previous works, the magnitude vectors can detect boundaries of a metric space, with boundary points having a larger contribution to magnitude (Bunch et al., 2021).

In this work, we seek to advance the research about magnitude in machine learning by addressing these challenges. Since the metric space structure of an entire dataset may not be the most useful application in machine learning, we instead therefore consider the magnitude of each individual data point by endowing each of them with a metric space structure and explore its properties. In particular, because of their ubiquity and the large dimensionality, we focus our analysis on image datasets, where each individual image is seen as a metric space. We then extend previous results by investigating the theoretical properties associated with magnitude vector in such a context. In addition, we study the properties of the magnitude vector for improving the adversarial robustness of existing neural network architectures for image classification. We propose a novel, fully differentiable, *magnitude layer*, which can serve as an input layer for any deep learning architecture. We show that it results in more robustness for several types of adversarial attacks, with an acceptable reduction in classification performance, paving the way for a new exciting direction in the creation of robust neural architectures. Moreover, since naive magnitude calculations may be computationally prohibitive for large images, we introduce a new algorithm that dramatically speeds up the computation of the magnitude vector. Leveraging the regular structure of images, this allows to conveniently approximate magnitude vectors for large images with minimal error. Intractable computational runtime often stymies the applicability of magnitude in machine learning and therefore hinders further the research of it; our algorithm opens

the door to using magnitude efficiently in machine learning research. Equipped with the speed up algorithm, we showcase possible use cases of magnitude vectors in machine learning in the realm of in edge detection and adversarial robustness.

Our contributions are summarized as follows:

- We formalize the notion of magnitude vectors for images and investigate the impact of the choice of different metrics.
- We introduce an algorithm that dramatically speeds up the computation of magnitude with little loss, which removes a critical roadblock in using magnitude vectors in machine learning applications.
- We provide a theoretical framework to understand the edge detection capability of magnitude vectors and report empirical supporting evidence.
- We demonstrate the capabilities of a novel, fully differentiable, *magnitude layer* for improving adversarial robustness of computer vision architectures.

## 2 THEORY

In this section, we introduce essential notions of the theory of magnitude and magnitude vectors and develop the theoretical background that will help interpreting the empirical results.

### 2.1 MATHEMATICAL BACKGROUND

We start by formally introducing the notion of a finite metric space.

**Definition 1.** *A metric space is an ordered pair* $(B, d)$*, where* $B$ *is a finite set and* $d$ *is a metric on* $B$*. We denote the cardinality of* $B$ *by* $|B|$*.*

In our application the set $B$ is a set of vectors $B \subset \mathbb{R}^n$ and the metric considered will be the $\ell_p$ norm. In order to define the magnitude of such a space we first define the similarity matrix.

**Definition 2.** *Given a finite metric space* $M = (B, d)$*, its similarity matrix is* $\zeta_M$ *with entries* $\zeta_M(i, j) = e^{-d(B_i, B_j)}$ *for* $B_i, B_j \in B$*.*

The similarity matrix can be seen as the weights arising from an exponential kernel. We are now in a position to define the magnitude vector and the magnitude of a finite vector space.

**Definition 3.** *Given a finite metric space* $M = (B, d)$ *with* $|B| = n$ *and similarity matrix* $\zeta_M$ *with inverse* $\zeta_M^{-1}$*, the magnitude vector of element* $B_i$ *is given by* $w_i = \sum_{j=1}^n \zeta_M^{-1}(i, j)$*. Moreover, the magnitude of* $M$*,* $mag_M$ *is* $\sum_{i,j=1}^n \zeta_M^{-1}(i, j) = \sum_{i=1}^n w_i$*.*

Not every finite metric space has a magnitude. In particular, the magnitude is not defined when the similarity matrix is not invertible; the magnitude therefore characterizes the structure of a metric space to some extent. It should be also noted that the definition of the magnitude vector is reminiscent of optimising a support vector machine. This connection has been pointed out for the Euclidean norm by Bunch et al. (2021).

### 2.2 THEORETICAL RESULTS

#### 2.2.1 THE MAGNITUDE OF AN IMAGE

This work focuses on the analysis of magnitude on images, by considering each individual image as its own metric space. We then first define how we build such a metric space from an image and then prove the existence of a magnitude on images.

We refer to the metric spaces on images as *image metric spaces* and define them as follows.

**Definition 4.** *Let* $I \in \mathbb{R}^{c \times n \times m}$ *be an image with* $c$ *channels, height* $n$ *and width* $m$*. Let* $\{V : \boldsymbol{v}_{ij}, i \in 1, \ldots, m; j \in 1, \ldots, n\}$ *be the set of* $c$*-dimensional vectors of pixel values in the image. Then the* image metric space $M(B, d)$ *is given by a set of vectors* $B \subset \mathbb{R}^{c+2}$ *of the form* $B = \{(i, j, v_{ij}^{(1)}, \ldots, v_{ij}^{(c)})^T : \forall \boldsymbol{v}_{ij} \in V\}$ *together with a metric* $d$ *on* $\mathbb{R}^{c+2}$*.*

Informally, we put all vectors corresponding to pixel values on a grid, concatenating the position vector with the pixel value vector and use the resulting vectors as the ground set $B$ for our finite metric space. Let us note that $|B| = m \times n$ and therefore the number of points in the metric space can be quite large even for moderately-sized images, a potential limitation that we address in Section 2.3.

We now turn to investigating when an image metric space has magnitude and, by extension, a magnitude vector. This is *a priori* not clear since, as we saw, the existence of magnitude depends on the properties of the metric space.

From the definition of magnitude (Definition 3), it is readily seen that an image metric space $M$ has magnitude if and only if its similarity matrix $\xi_M$ is invertible. As generic $n \times n$ matrices are invertible (i.e. subjecting any non-invertible matrix to a random perturbation will almost certainly result in an invertible matrix), we conclude that generic image metric spaces have magnitude. In fact, since the vectors $\boldsymbol{b} \in B$ of the image metric space by a factor $t > 0$ can be rescaled, we can define scaled metrics $d(\cdot, \cdot) \mapsto td(\cdot, \cdot)$. This rescaled metric space has magnitude except for finitely many $t > 0$ (Leinster et al., 2017, Proposition 2.8). In other words, we can always find a scaling which gives rise to a magnitude vector. The univariate function defined by this scaling is called the *magnitude function*.

In general, we are interested in computing the magnitude of all images in a whole dataset, and not only of a single image, such that we can compare their magnitude (vectors). Therefore, we would like theoretical guarantees that there exists a scaling such that every image in a dataset has a magnitude vector.

**Proposition 5.** *Let $M(B, d)$ be an image metric space. If $d$ is an $\ell_p$ norm with $0 < p < 2$, then every image metric space $M$ has magnitude.*

*Proof.* This is a special case of (Meckes, 2013, Theorem 3.6). $\qquad\square$

The above proposition theoretically only applies to $\ell_p$ norm with $0 < p < 2$. However, in practice, we find that on the various datasets we considered in this work, an $\ell_p$ norm with $p = 4, 10$ and the Hamming distance also lead to invertible similarity matrices, and thus to the existence of magnitude.

The image metric space exhibits substantial structure; in particular, there is an underlying regular subspace (the grid). To quantify this further, we use the notion of a product space.

**Lemma 6.** *Let $M_1(B_1, d_1)$ and $M_2(B_2, d_2)$ be two finite metric spaces with $d_1, d_2$ being $\ell_p$ norms. Its product metric space $M = M_1 \times M_2$ is a metric space with a metric $\rho_p : (B_1 \times B_2) \times (B_1 \times B_2) \to \mathbb{R}$ such that*

$$\rho_p((x_1 \times y_1), (x_2 \times y_2)) = \sqrt[p]{(d_1(x_1, x_2)^p + d_2(y_1, y_2)^p}, \quad 1 \leq p \leq \infty.$$

*Proof.* This is a special case of (Dovgoshey & Martio, 2009). $\qquad\square$

Note that the product space formulation gives a lot of freedom, since every positively *weighted* combination of $d_1$ and $d_2$ is also a metric on the product space.

**Magnitude vector on images based on harmonic analysis**   We now present our main result of this subsection which is an interpretation of the magnitude vector for images based on harmonic analysis. For this we consider grey scale images, however, our reasoning generalises to colour images. First, notice that a grey scale image can be seen as discretisation of a surface in $\mathbb{R}^3$, i.e. $z = f(x, y)$, where $x, y$ are the pixel positions and $z$ is the brightness. In general, it is not clear what an "outlier" on a continuous surface or curve is. In this paper we define an outlier as a point on the surface where the gradient is large, i.e. neighbouring points are further away w.r.t. some distance measure. In the image case, outliers are points where the brightness value is substantially different between neighbouring pixels. We can use this reasoning to define a filtration $\mathcal{F}$ on the vectors of $B$, $\boldsymbol{b}_{ij} = (i, j, v_{ij}^{(1)})^T$, namely $\emptyset \subseteq B^{(1)} \subseteq B^{(2)} \subseteq \cdots \subseteq B^{(K)} = B$ such that $\boldsymbol{v}_{ij}^{(1)} < \delta^{(k)}$ for $k = 1, \ldots, K$, where the $\delta^{(k)}$ are different brightness thresholds. Due to symmetry in the problem we also define $\mathcal{F}'$ with criterion $v_{ij}^{(1)} \geq \delta^{(k)}$. Further, we define the projection onto $\mathbb{R}^2$, $p : \mathbb{R}^3 \to \mathbb{R}^2$, $p(\boldsymbol{b}_{ij}) \mapsto (i, j)$. By

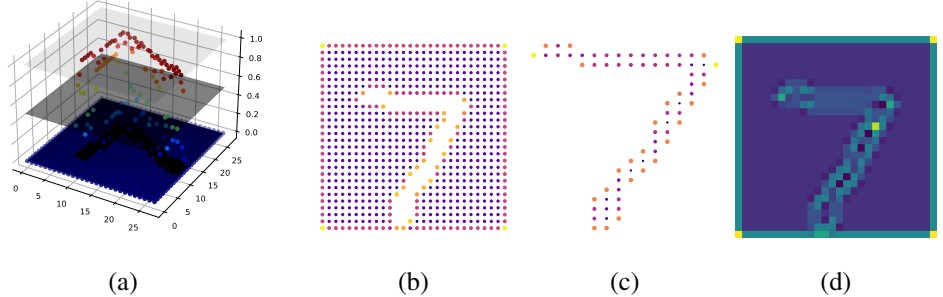

|     |     |     |     |
| --- | --- | --- | --- |
| (a) | (b) | (c) | (d) |

Figure 1: A schematic example of the filtration. In (a) we introduce three thresholds, $0, 0.5, 1$. In $\mathcal{F}$ all points below the grey planes are projected to the unit square and in $\mathcal{F}'$ all points above are projected. (b) is an illustration of the $0.5$ level of the filtration $\mathcal{F}$ and (c) is is the same for $\mathcal{F}'$. Point size and colour indicate the magnitude vector values. In (d) we reconstruct the magnitude vector.

considering the projections of each subset of $\mathcal{F}$ ($\mathcal{F}'$) we break the problem down into successive boundary detection of compact subsets of $\mathbb{R}^2$. To extend this reasoning to colour images we can either consider each channel as a grey scale image or define multi-dimensional filtrations. A visual description of our reasoning is found in Figure 1.

To investigate the effect of different metrics on boundary detection, we consider the behaviour of the weighting vector on the 2-dimensional grid. We closely follow the argument of Bunch et al. (2021), extending their results to the product space metric in the case when $d_i$ are $\ell_p$ norms with $p = 1, 2$ and

$$\rho((\boldsymbol{x}_1 \times z_1), (\boldsymbol{x}_2 \times z_2)) = \alpha_1 d_1(\boldsymbol{x}_1, \boldsymbol{x}_2) + \alpha_2 d_2(z_1, z_2) \tag{1}$$

for some positive weights $\alpha_1, \alpha_2$. For image we can consider the $\boldsymbol{x}_i$ as two-dimensional vectors indicating the position on the unit square and $z_i$ as the corresponding brightness value. Consider a regular 2d grid and the equation defining a weighting on the grid points $\zeta_M \boldsymbol{v} = (1, \ldots, 1)^T$. Using a continuous analogue, this can be written as a convolution (Bunch et al., 2021)

$$f \star v(\boldsymbol{x}) = \int_{\mathbb{R}^2} f(\boldsymbol{x} - \boldsymbol{y}) v(\boldsymbol{y}) = \int_{\mathbb{R}^2} e^{-\alpha d(\boldsymbol{x}, \boldsymbol{y})} v(\boldsymbol{y}) = \mathbb{I}_{[0,1]^2}(\boldsymbol{x}), \tag{2}$$

where $\mathbb{I}_{[0,1]^2}$ is the indicator function and $d$ is any translation invariant metric. Using the Fourier transform,

$$\mathrm{F}(f)(\boldsymbol{\xi}) = \int_{\mathbb{R}^2} e^{-i2\pi \boldsymbol{x} \cdot \boldsymbol{\xi}} f(\boldsymbol{x}) d\boldsymbol{x}, \tag{3}$$

its well-known properties Folland (1999) and the convolution theorem, we can derive an intuitive understanding of the magnitude vector and the effects of specific metrics. In the case of $d(\cdot, \cdot)$ being the Euclidean ($\ell_2$) norm, it has been shown in (Bunch et al., 2021) that

$$\frac{\Gamma(\frac{3}{2})}{\alpha \pi^{\frac{3}{2}}} (\alpha^2 - \sum_{i=1}^{2} \partial_i^2)^{\frac{3}{2}} \mathbb{I}_{[0,1]^2} = v, \tag{4}$$

where we generalised the result of Bunch et al. (2021) to the unit square. We can interpret equation 4 as the weighting $v$ being constant in the interior of the unit square ($\partial_i^2$ is zero in interior) and different to this constant on the boundary. In fact, the weighting will also be constant on the edges and corners.

In the Euclidean case it has been shown (Bunch et al., 2021) that the intuitive understanding translates rigorously to compact subsets of $\mathbb{R}^n$ with $n$ odd using distribution theory. Guided by this, we continue our argument using the Fourier transform.

If $d(\cdot, \cdot)$ is the $\ell_1$ norm, i.e. the Manhattan or Cityblock distance, we obtain a similar result to equation 4,

$$\left[ \prod_{i=1}^{2} (\alpha^2 - \partial_i^2) \right] \mathbb{I}_{[0,1]^2} = v, \tag{5}$$

in other words, the $\ell_1$ norm admits the same interpretation as $\ell_2$, however, this time the differential operators acts multiplicatively on each dimension.

For the product space metric $\rho(\cdot, \cdot) = \alpha_1 d_1(\cdot, \cdot) + \alpha_2 d_2(\cdot, \cdot)$ and $d_i$ are $\ell_p$ with $p = 1, 2$ it follows that the Fourier transform is a product of the single-metric results. Equipped with this insight, we can now explain the edge detection capabilities of the magnitude vector.

Consider a filtration $\mathcal{F}$ (and $\mathcal{F}'$) on a grey scale image and choose a subset of vectors $B^{(i)}$ from the filtration. Apply the projection map $p$ to every vector in this subset. The transformed set is a grid with potentially "missing" grid points on the domain $[0, n] \times [0, m]$. From the results on the unit square and the discretising of the $\partial$ operator, we expect a constant weighting vector except on the boundaries of the grid, i.e. on points adjacent to the missing grid points and on the boundaries of the domain. This procedure can be performed on every $B^{(i)} \in \mathcal{F}$ and the final magnitude vector can be seen as a a combination of the weightings of each step (see Figure 1). We find that our expectations from the theoretical result agree well with empirical results (see Appendix D). Moreover, we empirically find that Hamming distance and $\ell_p$ norms for $p \neq 1, 2$ have similar properties.

### 2.3 COMPUTATIONAL CONSIDERATIONS AND SPEED UP

As can be seen from Definition 3, the computation of the magnitude vector is achieved via a matrix inversion. This is a fundamentally expensive computation since for an $m \times n$ image, we need to invert an $N \times N$ square matrix with $N = mn$. In other words, even for moderately sized images there is a considerable computational challenge to overcome.

The authors of Bunch et al. (2021) propose an iterative algorithm based on the a small subspace of the finite metric space $M' \subset M$. Suppose the magnitude vector of $M'$ is known, then, when adding a single extra point to $M'$, the computation of the updated magnitude vector reduces to a simple matrix-vector multiplication. Empirically, we find that even for MNIST images this computation is considerably slower than the naive implementation due to the large number of steps in the algorithm. Furthermore, the full distance matrix of the image needs to be stored, which can quickly exceed memory.

When considering the metric space of images, we also have one fundamental advantage over methods, namely a grid structure. While every pixel in an image has a weighting, every weighting of a pixel can also be associated to a position on a two-dimensional grid. Any algorithm which speeds up the computation of weighting vectors should also implicitly consider this additional structure. Therefore, we propose a divide-and-conquer procedure to be able to handle large images efficiently.

According to the theoretical considerations of Subsection 2.2, the magnitude vector is constant in the interior of the unit square and also constant along its edges and corners. From this analysis we gain one key insight, namely creating patches from the image would result in different magnitude vector values along the edges, but all other magnitude vector elements would remain the same. To counter the resulting "edge effects" of the patches, we can choose overlapping patches, calculate the magnitude vector of the patches, and crop the transformed sub-images. Finally, piecing the patches together, we obtain an approximation of the full magnitude vector. As shown in Subsection 3.1, this results in considerable speed up with manageable error. Furthermore, it allows us to even calculate the magnitude vector of high-resolution images.

If even faster *inference* of the magnitude vector is required, we can view the calculation as a special type of image-to-image translation problem and use current techniques available.

## 3 EXPERIMENTS

Given a theoretically-based algorithm to efficiently calculate the magnitude vector on large images, we can investigate its effect empirically. We turn now to our experiments which begin first with an analysis of the runtime using the speed up algorithm. Armed with a fast calculation, we are then free to explore potential applications of the magnitude vector in machine learning applications. We investigate the utility of magnitude vectors in two domains which magnitude was not originally designed for, namely in edge detection and adversarial robustness, and find that it can prove useful.

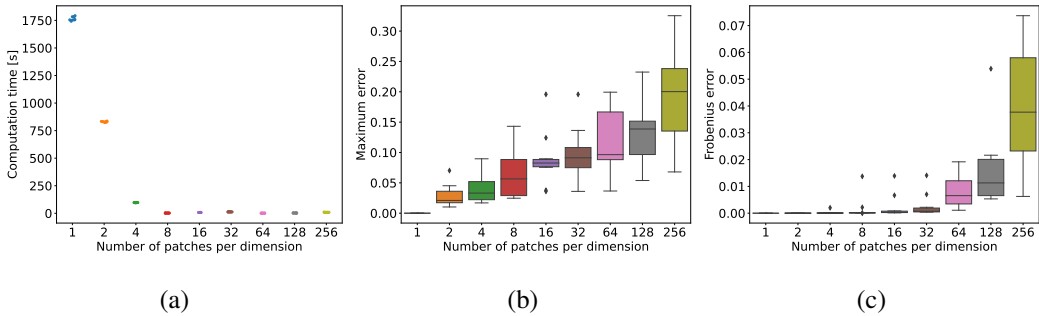

(a)                                 (b)                                 (c)

Figure 2: (a) The computation time in seconds of the magnitude vector calculation for each of the ten images. Even a small number of patches the computational time deceases drastically. (b) and (c) show box plots of the approximation errors. Although the maximum error can be quite large compared to the maximum difference of 1, the Fröbenius error remains small, indicating a good overall approximation.

## 3.1 FAST MAGNITUDE CALCULATION AND HIGH-RESOLUTION IMAGES

To illustrate the benefits of the patching method, we consider the first ten images of the NIH Chest X-ray dataset (Wang et al., 2017). The dataset consists of 100,000 de-identified X-ray (grey scale) images with a resolution of $1024 \times 1024$. Hence, a full-scale magnitude calculation would involve inverting a roughly $10^6 \times 10^6$ matrix which is far beyond current computational power.

To estimate the error induced by the patching algorithm, we rescale the images to a $256 \times 256$ resolution and calculate the magnitude via matrix inversion. Then, we divide the rescaled image into patches and repeat the calculation. We used the a $\ell_1$ norm and weights of 1 for both the grid points and intensity values. To ensure comparability we min-max scaled the magnitude vectors. The error is measured in maximum absolute deviation on a pixel level and also in terms of Fröbenius distance between the two magnitude vectors

$$\text{error} = \frac{\sum_i (\text{magnitude vector}_{\text{exact}} - \text{magnitude vector}_{\text{patches}})^2}{\sum_i (\text{magnitude vector}_{\text{exact}})^2}. \tag{6}$$

The results can be found in Figure 2. We observe a significant reduction (several order of magnitudes) when using our fast magnitude calculation method, with limited reconstruction error.

To illustrate the effect of the magnitude vector on full-scale images we used the patch-algorithm on the first image of the dataset (see Supplementary Figure 7). We also investigated effect of increasing the weight $\alpha_2$ of the pixel values in the product space metric and the results can be found in the Supplementary Material.

## 3.2 MAGNITUDE VECTORS AND EDGE DETECTION

Now that we have a computationally efficient method to calculate magnitude, we turn towards the first of two novel applications of magnitude: developing a proof-of-concept of the capabilities of magnitude for edge detection. Indeed, as shown in Section 2, the magnitude vector is expected to be higher on edges present in the images. This leads naturally to an edge detection algorithm where pixels whose magnitude vector values are larger than a threshold are classified as edges.

To assess the performance of a magnitude-based approach on this task, we compare it with the famous Canny detector (Canny, 1986), which has been widely used for edge detection in images. Because the definition of an edge is a subjective concept, we use the edges given by the Canny detector as a ground truth and tune the hyper-parameters of our magnitude-based edge detector such as to match the Canny edges as close as possible. We use the Sørensen-Dice coefficient, $\mathcal{D}$, to assess the similarity between both types of edges. Both algorithms can be seen as classifier on a pixel level operating on two classes ({ edge, no-edge }), thereby outputting a mask with same dimensions of the original image. The Sørensen-Dice coefficient between one predicted edge mask ($\text{Edge}_{\text{hat}}$) and the reference edge mask ($\text{Edge}_{\text{ref}}$) then writes:

Table 1: Sørensen-Dice coefficient on held-out test of the magnitude-based method with respect to the Canny method.

| FASHIONMNIST | CIFAR | X-ray |
|---|---|---|
| 0.59±0.12 | 0.61±0.07 | 0.33±0.07 |

$$\mathcal{D}(\text{Edge}_{\text{hat}}, \text{Edge}_{\text{ref}}) = \frac{2\text{TP}}{2\text{TP} + \text{FP} + \text{FN}}$$

where $\text{TP}, \text{FP}$ and $\text{FN}$ stands for the number True Positive, False Positive and False Negative pixels predictions respectively.

We then use Bayesian optimization of the Sørensen-Dice coefficient (Sorensen, 1948; Dice, 1945) on a training set to find the best hyperparameters of the magnitude-based edge detector. In particular, we tune the threshold at which magnitude vectors predict for an edge and the $\alpha_2$ from Equation 1. We then report the metric on an hold-out test set, and the results are shown in Table 1. We ran this experiment on three different datasets : FashionMNIST, CIFAR and an X-ray image dataset. For the X-ray dataset, we use the *patched* version as described in Section 2.3. On Figure 3, we show examples from the held-out test set on the Fashion MNIST dataset. In Appendix A, we show examples from the two other datasets. A visual inspection shows that edges can be recovered quite accurately. However, the edges masks obtained through magnitude appear more noisy.

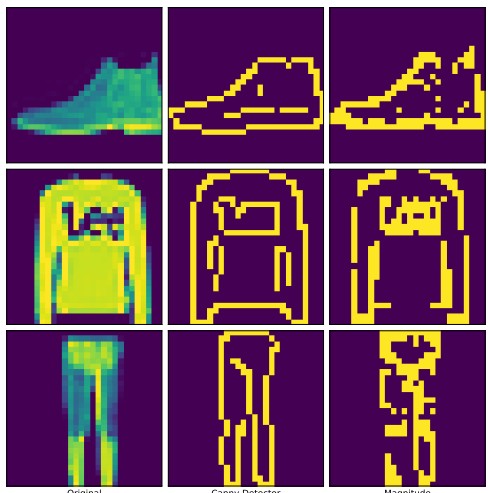

Figure 3: Examples of edge detection on the FashionMNIST dataset. The left-most column show the initial images, the center shows the edges mask obtained with the Canny detector, the right-most column shows the edges mask obtained with the magnitude-based approach.

### 3.3 MAGNITUDE VECTORS AND ADVERSARIAL ROBUSTNESS

As we saw in Subsection 3.2 and Supplementary Figure 6, the magnitude calculation transforms an original image into a representation of its edges. We now want to explore another facet of its utility in machine learning by investigating its potential use for improving adversarial robustness. In the following, the datasets we use are MNIST, KMNIST and FashionMNIST.

The magnitude layer is a zero-parameter feature transformation consisting, sequentially, in

1. the computation of the distance matrix,

2. the exponentiation of the negative distance matrix element-wise,

3. the inversion the resulting matrix,

4. the summing of the resulting matrix over its rows.

Crucially, this layer can be used as an input layer to any image-classification network. While the transformation is still, in principle, differentiable, we are presented with numerous opportunities to introduce a step function into the calculation. One possibility is inspired by feature squeezing (Xu et al., 2017), which quantizes the pixel values. In the magnitude calculation we can differentiably quantize the values of the inverse similarity matrix just before the summation in step 4. This step function ensures that the gradients with respect to the input data are zero and, therefore, *any* white-box attacks which rely on this gradient (such as FGSM, PGD, Carlini-Wagner) must fail.

While the introduction of the step function guarantees trivial "robustness", we also investigate the efficacy of transferred adversarial examples. To this end, we train two LeNet models (LeCun et al., 1989) for 30 epochs each on our datasets, one with a magnitude input layer, one without (the base model). We then generate adversarial examples w.r.t. the base model and test them on the model with magnitude. Our results for an FGSM attack and various quantization levels are summarised in Table 2. We note that, although our model shows the best performance in terms of adversarial evasion, we emphasise that these results are not directly comparable. While simple FGSM attacks fail altogether on the squeezed magnitude layer due to the step function, we calculated robustness via a transfer of adversarials from a conventional LeNet model. The feature squeezing setup is similar to the original setup from Xu et al. (2017), however, we used our own quantization layer which divides the features into equally spaced levels and the models were trained on the squeezed images. The base model is a simple undefended LeNet. We also investigate the robustness properties of a non-quantized magnitude input layer which $\ell_1$ norm, which shows substantially increased robustness over the baselines of the undefended model and for large $\epsilon$ feature squeezing.

The number of levels in the feature squeezing was chosen to be small due to the fact that there are large brightness in the datasets we have mainly white objects on black background and, therefore, a large compression would be expected to give the largest robustness. The levels of the magnitude vector were chosen to be larger as the models performed worse with smaller levels.

| | $\epsilon$ | Levels | | | Base | Squeeze Levels | | | Standalone $\ell_1$ |
|---|---|---|---|---|---|---|---|---|---|
| | | 50 | 100 | 1000 | | 2 | 3 | 5 | |
| MNIST | 0 | 97.79 | 98.05 | 97.63 | 99.01 | 98.57 | 98.68 | 98.73 | 98.04 |
| | 0.05 | 71.90 | 72.88 | 68.23 | 94.78 | 100.0 | 100.0 | 100.0 | 90.78 |
| | 0.1 | 54.50 | 65.41 | 62.02 | 81.57 | 100.0 | 100.0 | 100.0 | 81.36 |
| | 0.2 | 54.95 | 67.24 | 60.65 | 42.10 | 100.0 | 100.0 | 32.20 | 65.68 |
| | 0.3 | 54.84 | 68.38 | 60.39 | 21.76 | 100.0 | 3.47 | 32.20 | 51.55 |
| | 0.4 | 46.14 | 61.56 | 54.57 | 14.37 | 100.0 | 3.47 | 2.53 | 39.77 |
| | 0.5 | 28.00 | 43.87 | 39.34 | 11.95 | 18.43 | 3.47 | 2.53 | 33.65 |
| | 0.6 | 25.66 | 41.77 | 38.19 | 11.72 | 6.26 | 3.47 | 2.53 | 28.82 |
| | 0.7 | 23.56 | 39.12 | 35.93 | 12.95 | 6.26 | 3.47 | 2.24 | 21.36 |
| KMNIST | 0 | 81.23 | 83.75 | 80.26 | 88.06 | 87.60 | 87.84 | 88.25 | 83.26 |
| | 0.05 | 53.91 | 64.27 | 53.25 | 78.34 | 100.0 | 100.0 | 100.0 | 70.77 |
| | 0.1 | 32.52 | 42.15 | 30.29 | 55.54 | 100.0 | 100.0 | 100.0 | 55.85 |
| | 0.2 | 27.05 | 34.93 | 26.14 | 18.69 | 100.0 | 100.0 | 19.88 | 41.64 |
| | 0.3 | 27.94 | 36.94 | 28.20 | 6.56 | 100.0 | 3.89 | 19.88 | 35.44 |
| | 0.4 | 24.60 | 31.7 | 24.77 | 3.97 | 100.0 | 3.89 | 2.31 | 31.22 |
| | 0.5 | 16.61 | 22.27 | 16.43 | 3.68 | 19.42 | 3.89 | 2.31 | 28.74 |
| | 0.6 | 16.52 | 21.74 | 16.51 | 3.85 | 5.31 | 3.89 | 2.31 | 25.26 |
| | 0.7 | 16.24 | 20.94 | 16.39 | 4.03 | 5.31 | 3.89 | 1.90 | 20.85 |
| FashionMNIST | 0 | 81.45 | 82.49 | 80.46 | 89.0 | 82.82 | 82.88 | 82.82 | 86.02 |
| | 0.05 | 77.42 | 75.53 | 74.67 | 24.79 | 100.0 | 100.0 | 100.0 | 55.43 |
| | 0.1 | 56.85 | 61.48 | 66.17 | 10.70 | 100.0 | 100.0 | 100.0 | 43.39 |
| | 0.2 | 48.86 | 53.89 | 60.34 | 5.10 | 100.0 | 100.0 | 23.68 | 35.24 |
| | 0.3 | 44.13 | 48.83 | 55.95 | 1.82 | 100.0 | 3.37 | 23.68 | 30.05 |
| | 0.4 | 38.15 | 42.66 | 51.77 | 1.21 | 100.0 | 3.37 | 4.76 | 24.81 |
| | 0.5 | 27.30 | 33.13 | 43.29 | 1.42 | 23.26 | 3.37 | 4.76 | 20.45 |
| | 0.6 | 28.39 | 31.28 | 42.00 | 1.55 | 4.47 | 3.37 | 4.76 | 16.22 |
| | 0.7 | 24.78 | 29.53 | 39.88 | 1.72 | 4.47 | 3.37 | 2.54 | 13.54 |

Table 2: The accuracies of the attacked networks. From left to right: quantized magnitude layer with different quantization levels, the undefended model, the model with feature squeezing and various levels, and the magnitude layer as an input layer. The zero epsilon values are the base accuries. For the non-zero epsilon, the accuracy is the ratio of non-adversarial examples to base accuracies.

## 4 RELATED WORK

An early version of magnitude has been introduced by Solow & Polasky (1994) where it was defined as an "effective number of species", however with little mathematical development. The subject has been picked up formally by Leinster (2010) where connection to theory of enriched categories has been made. From there, the magnitude of many mathematical objects has been studied, such as spheres (Willerton, 2014), odd balls (Willerton, 2018; 2017), compact sets (Meckes, 2015; Leinster & Willerton, 2013), convex bodies (Meckes, 2020), and graphs (Leinster, 2019).

As stated in the introduction, magnitude and magnitude vectors have not been studied from a machine learning perspective except for the work of Bunch et al. (2021) that analysed the capability of magnitude vectors for boundary detection. In contrast, our work extends the results from this study to images and provide the necessary theoretical grounds. To the best of our knowledge, this represents the first work exploring magnitude vectors for images. In particular, using magnitude vectors for adversarial robustness has never been explored before.

## 5 DISCUSSION

This paper investigated the magnitude vector of images. In the first part, we stated several theoretical properties of the magnitude vector on images and showed how the outlier detection property translates to edge detection in images. Although the theoretical part is based on analogies, rather than rigorous mathematical proofs, we can still distil the essence of the behaviour of the magnitude vector. In particular, these theoretical insights are confirmed by practical experiments. We also propose an algorithm for an approximate calculation of the magnitude vector leading to a significant speed up of computations. While this works well in practice, as shown in Subsection 3.1, the exact computation of magnitude vector on a metric space with a large number of points still remains an open problem.

Our algorithm to speed up the computation allows magnitude to be used more generally as the computational bottleneck of its naive computation can be bypassed. Investigating the capabilities of magnitude vectors in machine learning applications, we studied two application areas in more depth, namely edge detection and adversarial robustness, showing promising and overall favourable results in both cases.

As for edge detection, we compared a magnitude-based approached with the well known Canny edge detector. We show that the overlap between the edges found by the Canny detector and the magnitude vector is noteworthy, though the magnitude vector can appear a bit noisier. In particular, for low resolution colour images (*e.g.* CIFAR-10), the edges are very noisy and a thorough investigation for the magnitude vector on colour images should be conducted. This is nonetheless impressive, since the magnitude vector is a general-purpose image transformation, unlike the Canny detector, and therefore shows an exciting potential for improving exisiting edge detection tools.

We also considered a potential application of the magnitude vector as a simple mean of defence against adversarial attacks. We first noted that a step function such as a quantization function can be introduced into the magnitude calculation, providing a trivial but effective defence against any white box attack. We went further by investigating the property of transferred adversarial examples. Our model shows reasonable robustness and performs better than the baseline method of feature squeezing. With these encouraging results we aim to investigate the use of magnitude in adversarial evasion and detection in future work.

## REPRODUCIBILITY STATEMENT

We provide all code to create figures and results in this paper in the supplementary material. All datasets are publicly available and downloadable via the code provided.

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

## A   SUPPLEMENTARY ILLUSTRATIONS OF THE EDGE DETECTION CAPABILITIES OF THE MAGNITUDE VECTORS.

On Figures 4 and 5, we show examples of edge detection from the test sets of the CIFAR and X-ray datasets respectively.

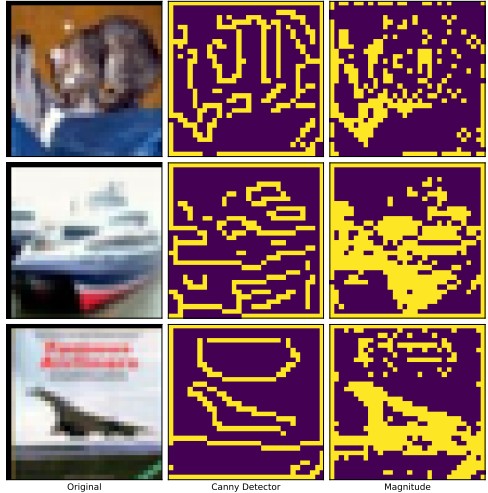

Figure 4: Three randomly selected examples of edge detection on the CIFAR dataset. Left left-most column show the initial images, the center shows the edges mask obtained with the Canny detector, the right-most column shows the edges mask obtained with the magnitude-based approach.

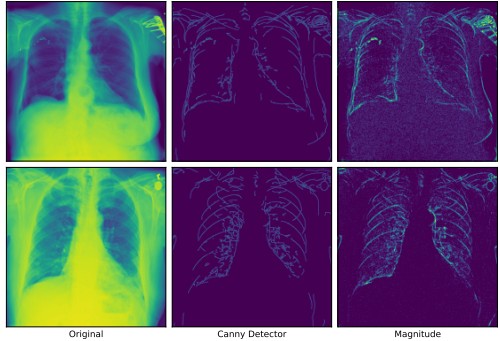

Figure 5: Two randomly selected examples of edge detection on the X-ray dataset. Left left-most column show the initial images, the center shows the edges mask obtained with the Canny detector, the right-most column shows the edges mask obtained with the magnitude-based approach.

## B  MAGNITUDE VECTORS WITH DIFFERENT METRICS

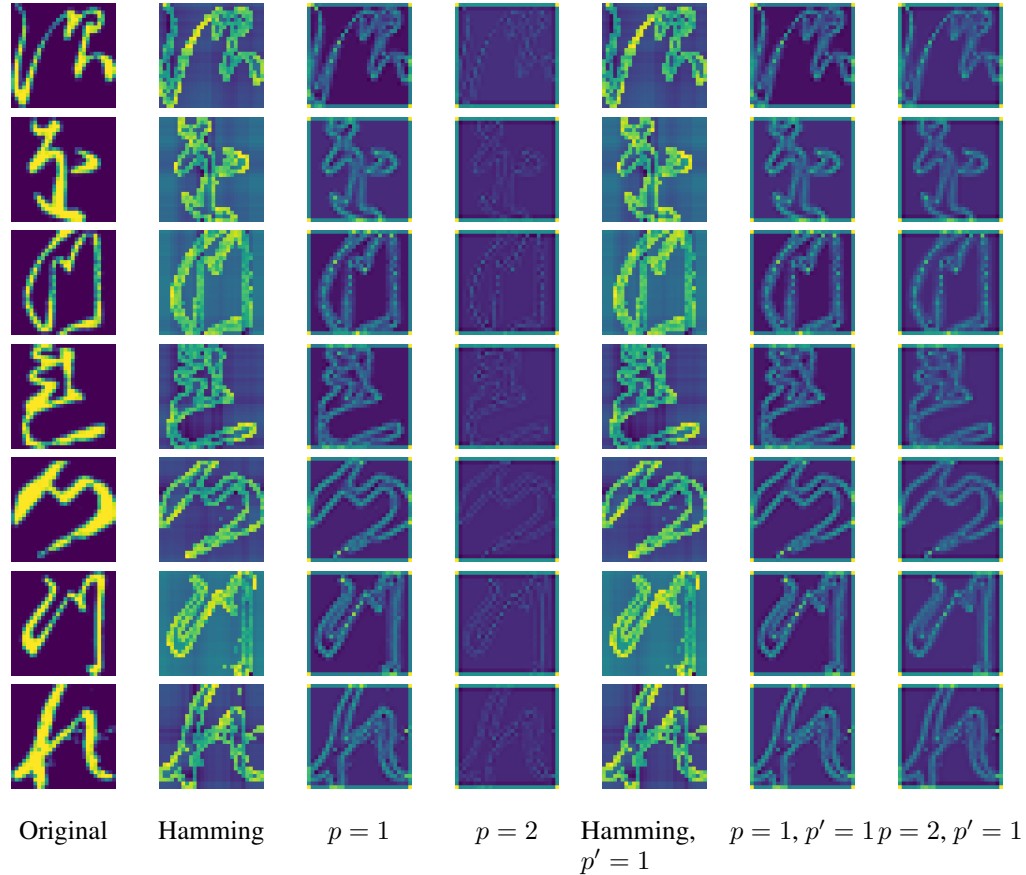

| Original | Hamming | $p = 1$ | $p = 2$ | Hamming, $p' = 1$ | $p = 1, p' = 1$ | $p = 2, p' = 1$ |

Figure 6: A comparison of the effect of different metrics in the magnitude calculation on KMNIST. The first column is the original image and the following three columns are the magnitude vectors with different $\ell_p$ metrics. The last three columns use a product space metric with $\alpha_1 = \alpha = 2 = 1$, grid metric $\ell_p$ and brightness metric $\ell_{p'}$.

## C  THE MAGNITUDE OF FULL SCALE IMAGES

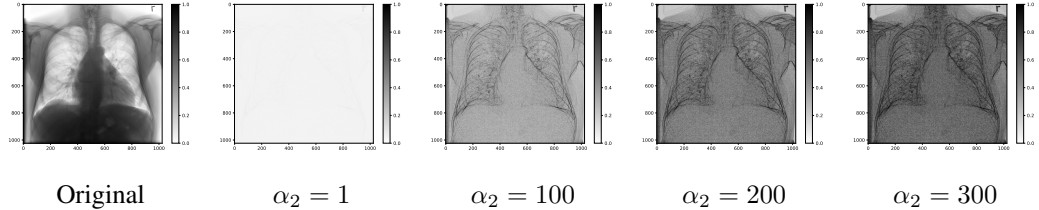

| Original | $\alpha_2 = 1$ | $\alpha_2 = 100$ | $\alpha_2 = 200$ | $\alpha_2 = 300$ |

Figure 7: An illustration of the effect of $\alpha_2$ on the magnitude calculation of full-scale images.

## D  COMPARISON OF THE FILTRATION AND THE FULL MAGNITUDE

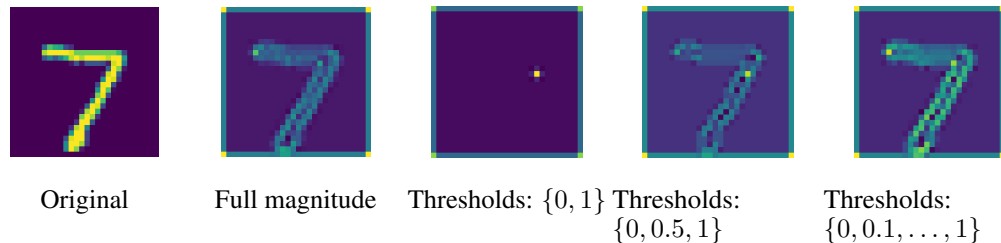

| Original | Full magnitude | Thresholds: $\{0, 1\}$ | Thresholds: $\{0, 0.5, 1\}$ | Thresholds: $\{0, 0.1, \dots, 1\}$ |

Figure 8: An illustration of the magnitude based on filtrations with metric $\ell_1$.

