# OpenReview forum: "The magnitude vector of images"
_ICLR.cc/2022/Conference — ICLR 2022 Submitted_

### Official Review · Reviewer_cxFw · 2021-10-25

**Correctness:** 3
**Technical Novelty And Significance:** 1
**Empirical Novelty And Significance:** 2
**Recommendation:** 3
**Confidence:** 4

**Main Review:**

This paper does not provide any theoretically-backed reason why magnitude vectors of images should be considered, and in fact the given definition of a vector space associated with an image appears quite dubious: the existence of the magnitude of an image depends on individual brighness values, which are irrelevant to the actual overall meaning of the image; in addition, the metric mixes the position of the pixels with their brightness, which looks very unnatural.

The algorithm to speed up the computation of magnitude vectors is also not theoretically backed in any way, and only supported by empirical experiments.

Finally, the two given application of magnitude vectors to computer vision are very weak. In edge detection, it is shown that with appropriate hyperparameter tuning, magnitude vectors can recover a noisy version of the output of the Canny detector. The authors never observe that edge detection is a local problem, whereas the magnitude vector is a global quantity; this appears to me as an intrinsic theoretical limitation to the application of magnitude vectors to edge detection. As for adversarial robustness, it seems quite obvious that generated adversarial examples for a base LeNet are classified more correctly by a modified LeNet (regardless of how the network is actually modified).

Here are some additional minor comments:
- After Definition 1: a *finite* set of vectors
- Lemma 6: this is obvious, there is no need to cite a 2009 preprint


**Summary Of The Paper:**

This paper applies the machinery of magnitude vectors of metric spaces to extract information from individual images (which can be themselves be seen as finite metric spaces having one point for each pixel). After some theoretical considerations, the paper proposes a divide-and-conquer algorithm to approximate the magnitude vector of an image which is significantly faster than the naive inversion of a $N\times N$ matrix (where $N$ is the number of pixels in the image). Finally, the magnitude vector is shown to correlate with the result of traditional edge detectors (after hyper-parameter tuning) and an application to adversarial robustness is described.

**Summary Of The Review:**

This paper presents an application of magnitude vectors to image, which is not convincing both from a theoretical perspective (why should the given definition be useful? why should the given speed-up algorithm work well?) and from an empirical perspective (the results on edge detection and adversarial robustness are weak).

---

### Official Review · Reviewer_f2ou · 2021-11-01

**Correctness:** 3
**Technical Novelty And Significance:** 3
**Empirical Novelty And Significance:** 3
**Recommendation:** 3
**Confidence:** 3

**Main Review:**

Strengths:
- The magnitude of a metric space seems to have various interesting properties encoding other quantities of interest.
- The fast magnitude computation method is interesting and seems to be efficient although the scale of the significance of approximation error needs more discussion.

Weaknesses:
- There is a lack of motivation for considering the magnitude as a quantity for any purpose. My understanding from prior work is that the magnitude is a rather theoretical curiosity that “turns out to encode many invariants … including, volume, capacity, dimension and intrinsic volume” (Leinster 2017). But after reading this paper as well as prior work, I still do not have an intuition as to what it is expected to encode about an image. I will expand on this issue below under the experimental limitations.
- Theoretical contributions of the work are not clear. A few definitions, propositions and lemma are restated from prior work. Together they seem to provide expressions for computing the magnitude of a single image. However, all of these come from prior work.
- My understanding is that the contribution in the Section 2.2 is the particular definition of metric space for an image (Def. 4). The rest of the section applies prior tools to this metric space. There is a lack of justification for this definition. For example, why should one treat individual pixels as a point and why should we use spatial coordinates in the vector representation?
- The notation is confusing. For example, Def. 3 and Def. 1 use n to denote both the cardinality of B and the dimensionality of elements in B. Eq. 4 and 5 are also using notations with proper definition that comes from Bunch et al 2021.
- The significance of Section 3.2 is not clear. Are authors claiming “The magnitude of an image is as good as the Canny Edge detector.”? If so, can authors clarify why this is a significant observation? Given Figure 3, I’m not sure I would agree with this claim either, as the magnitude-based approach has mostly failed on the image of the pants.
- The adversarial robustness results in Section 3.3 need improvement considering major works on attacking discretization layers that achieve spurious robustness through gradient masking / gradient obfuscation. As tested by authors, black-box attacks are one approach, however, FGSM is not a particularly strong attack. Various other guidelines are suggested in [1,2].

[1] Athalye, Anish, Nicholas Carlini, and David Wagner. "Obfuscated gradients give a false sense of security: Circumventing defenses to adversarial examples." International conference on machine learning. PMLR, 2018.
[2] Florian Tramèr, Nicholas Carlini, Wieland Brendel, Aleksander Madry:
On Adaptive Attacks to Adversarial Example Defenses. NeurIPS 2020

**Summary Of The Paper:**

This paper explores the application of the quantity “magnitude of a metric space” for images. Based on prior work, magnitude is a quantity originating from category theory extended to metric spaces with surprising geometric properties (Leinster 2017). This paper defines the magnitude as a vector valued function of a metric space (Def. 3). Through defining an image as a metric space, the magnitude of a single image is defined (Def. 4). Authors then state conditions under which magnitude of an image is known to exist (Prop 5). Then the paper defines a filtration operation that computes the magnitude vector on multiple thresholded projections of the input. Section 2.3 introduces an approximate computation method for the magnitude and Section 3.1 evaluates the efficiency and error of this approximation. Section 3.2 applies the method for edge detection and compares it to Canny detector. Section 3.3 uses the magnitude for adversarial robustness and claims gains.

**Summary Of The Review:**

The paper requires major improvements to motivation, empirical results, and rigour of theoretical contributions.

---

### Official Review · Reviewer_qYUC · 2021-11-02

**Correctness:** 3
**Technical Novelty And Significance:** 2
**Empirical Novelty And Significance:** 2
**Recommendation:** 3
**Confidence:** 3

**Main Review:**

## Page 2

#### Definition 1
This should read "A finite metric space is an ordered pair..."

#### Definition 4
- The notation to define $V$ is nonstandard and unclear.
- The notation $M(B,d)$ is nonstandard: It should either be $M$ or $(B,d)$ unless $M$ some kind of mapping. I.e. $M=(B,d)=M(B,d) $ is not a consistent choice of notation.
- In the definition of $B$ (as well as $V$) it would be wise to avoid an unrestricted comprehension.

## Page 3

> As generic $n \times n$ matrices are invertible (i.e. subjecting any non-invertible matrix to a random perturbation will almost certainly result in an invertible matrix), we conclude that generic image metric spaces have magnitude.

- This is not a conclusion, it is an assumption.

> In fact, since the vectors b ∈ B of the image metric space by a factor t > 0 can be rescaled, we can define scaled metrics $d(·, ·) \mapsto td(·, ·)$.

- This only makes sense if $d$ is induced by a norm. The sentence, as written, does not make sense.

#### Lemma 6
- This is a standard metric space construction, and applies even more generally than its statement here. Its inclusion is inappropriate.

> We can use this reasoning to define a filtration $F$ on the vectors of B...
- Typo, should be $v_{ij}^{(k)}$ not $v_{ij}^{(1)}$.
- observe the se are just the sublevel sets of the function $f$, calling this a "filtration" is confusing.

##  Page 4
Can you please explain the purpose of the derivations and what the results are? It is unclear what here is new or novel and what is a result from Bunch et al. (2021).


## Pages 5 and 6
I would an explanation here of what the new algorithm is. Additionally if you are going to claim that your algorithm is significantly faster you need to show a baseline to compare the speed with others, e.g. Bunch et al. (2021). Furthermore why have you chosen an X-ray dataset as the only dataset to present benchmarks with?








**Summary Of The Paper:**

The authors consider a relatively new finite metric space quantity known as the "magnitude" and its applications within computer vision.

The paper proceeds with introducing and defining the magnitude quantity before exploring some key properties.  The contributes the authors outline are a study of
* relationship between 'outlier detection' and 'edge detection'
* introduction of 'magnitude layer' as an adversarial defence
* new algorithm to speed up the computation of the magnitude vect

**Summary Of The Review:**

The reason that I have recommended this paper are several fold.

Chiefly it is unclear what the novel contribution of the authors are in this paper. Taking the contributions one by one:
 - It is unclear what the benefit or novelty there is from the study of outliers and edge detection
- The improved algorithm is not explained, presented, or studied empirically with reference to any baseline.
- The new layer the authors present involves a computationally prohibitive matrix inversion that will limit its applicability.

---

### Official Review · Reviewer_WNxv · 2021-11-08

**Correctness:** 2
**Technical Novelty And Significance:** 2
**Empirical Novelty And Significance:** 2
**Recommendation:** 3
**Confidence:** 4

**Main Review:**

Strength:
The paper discuss a concept that is relatively new in the machine learning community. The magnitude of a metric space has been studied in topology, and this paper demonstrates that it can potentially be used in image-related ML problems.

Weakness:
Given the results in the current paper, I am not fully convinced that the concept of magnitude vector is really important for learning problems. To better demonstrate its importance, the authors should provide more comprehensive experimental results.

For edge detection: the authors only provided some visual examples for the Fashion MNIST dataset. From the three images that the authors provided in the paper, the results of magnitude-based edge detection is visually worse than Canny detector. The authors did not provide any quantitative results on the quality of edge detection. I think a more rigorous study and evaluation is needed to show its benefits.

For adversarial robustness, the authors only provided results on three relatively simple datasets: MNIST, KMNIST, Fashion MNIST, with a relatively weak attack (FGSM). To demonstrate the method of the proposed approach, the authors should provide results on more datasets and more baseline algorithms.

====
After response: the authors did not post specific response so I decided to keep my score.

**Summary Of The Paper:**

This paper discuss a concept of magnitude vector of images. The authors provided a fast algorithm for computing the magnitude vector, and discuss the potential applications in edge detection and adversarial robustness.

**Summary Of The Review:**

The experimental results are not strong enough to demonstrate the benefits of the concept of magnitude vector of images.

---

### Author Response · Authors · 2021-11-16
**Thank you for the comments**

We thank the reviewers for their time and helpful comments. We will implement them in an an updated version of the manuscript.

---

### Decision · Program_Chairs · 2022-01-20

**Decision:**

Reject

**Comment:**

This paper discusses a relatively new concept called "magnitude" for finite metric spaces and investigates its potential applications in machine learning, in particular for computer vision.

Reviewers generally agree that this is an interesting concept and appreciate the algorithm for reducing its computational cost.
However, there are concerns (1) that the concept is not well-motivated theoretically for machine learning problems
(2) the experimental results, for edge detection and adversarial robustness, are not convincing. More rigorous empirical work should be carried out.